# Association Mapping of Verticillium Wilt Disease in a Worldwide Collection of Cotton (*Gossypium hirsutum* L.)

**DOI:** 10.3390/plants10020306

**Published:** 2021-02-05

**Authors:** Adem Bardak, Sadettin Çelik, Oktay Erdoğan, Remzi Ekinci, Ziya Dumlupinar

**Affiliations:** 1Department of Agricultural Biotechnology, Faculty of Agriculture, Kahramanmaraş Sütçü İmam University, Kahramanmaraş 46100, Turkey; zdumlupinar@ksu.edu.tr; 2Department of Agricultural Biotechnology, Faculty of Agriculture, Bingol University, Bingol 12000, Turkey; sadettincelik@bingol.edu.tr; 3Department of Organic Farming Business Management, School of Applied Sciences, University of Pamukkale, Civril, Denizli 20600, Turkey; oktaye@pau.edu.tr; 4Department of Field Crops, Faculty of Agriculture, Dicle University, Diyarbakir 21280, Turkey; remzi.ekinci@dicle.edu.tr

**Keywords:** cotton, marker, SNP, GBS, association mapping, *Verticillium dahlia* Kleb., pathotype, stress genes

## Abstract

Cotton (*Gossypium* spp.) is the best plant fiber source in the world and provides the raw material for industry. Verticillium wilt caused by *Verticillium dahliae* Kleb. is accepted as a major disease of cotton production. The most practical way to deal with verticillium wilt is to develop resistant/tolerant varieties after cultural practices. One of the effective selections in plant breeding is the use of marker-assisted selection (MAS) via quantitative trait loci (QTL). Therefore, in this study, we aimed to discover the genetic markers associated with the disease. Through the association mapping analysis, common single nucleotide polymorphism (SNP) markers were obtained using 4730 SNP alleles. As a result, twenty-three markers were associated with defoliating (PYDV6 isolate) pathotype, twenty-one markers with non-defoliating (Vd11 isolate) pathotype, ten QTL with Disease Severity Index (DSI) of the leaves at the 50–60% boll opening period and eight markers were associated with DSI in the stem section. Some of the markers that show significant associations are located on protein coding genes such as protein Mpv17-like, 21 kDa protein-like, transcription factor MYB113-like, protein dehydration-induced 19 homolog 3-like, F-box protein CPR30-like, extracellular ribonuclease LE-like, putative E3 ubiquitin-protein ligase LIN, pentatricopeptide repeat-containing protein At3g62890-like, fructose-1,6-bisphosphatase, tubby-like F-box protein 8, endoglucanase 16-like, glucose-6-phosphate/phosphate translocator 2, metal tolerance protein 11-like, VAN3-binding protein-like, transformation/transcription domain-associated protein-like, pyruvate kinase isozyme A, ethylene-responsive transcription factor CRF2-like, molybdate transporter 2-like, IRK-interacting protein-like, glycosylphosphatidylinositol anchor attachment 1 protein, U3 small nucleolar RNA-associated protein 4-like, microtubule-associated protein futsch-like, transport and Golgi organization 2 homolog, splicing factor 3B subunit 3-like, mediator of RNA polymerase II transcription subunit 15a-like, putative ankyrin repeat protein, and protein networked 1D-like. It has been reported in previous studies that most of these genes are associated with biotic and abiotic stress factors. As a result, once validated, it would be possible to use the markers obtained in the study in Marker Assisted Selection (MAS) breeding.

## 1. Introduction

Cotton (*Gossypium* spp.) is an important plant that provides raw materials to the textile, food and feed industries [1]. Biotic and abiotic stress factors in cotton cause a decrease in yield and quality. As a biotic stress factor, Wilt disease is caused by *Verticillium dahliae* Kleb., which is known as the most disruptive and destructive one in the world [2,3,4]. The pathogen can cause wilting in more than 400 plant species other than cotton, such as vegetables, legumes, ornamental plants, industrial plants, fruit trees and weeds [5]. Today, worldwide estimated crop loss of Verticillium wilt is reported as 1.5 million bales [6]. It has been reported that Verticillium wilt caused approximately 148 thousand cotton bale losses in the 2010 production season in the USA [7] and 480 million bales from 1990–2014 [8].

*V. dahliae* first gradually enters the tissue from the root, settles into the xylem and begins to develop here, and causes clogging by sedimentation in the veins of the stem and causes chlorosis and necroses in the leaves, and then pallor forms thyllose [9,10]. The pathogen prevents the transfer of water and other mineral substances from roots to the leaves and tissues, and then starting from the lower leaves, it causes wilting, drying and a reduction in photosynthesis, changing yield and fiber quality characteristics and shedding in small bolls [11]. There are two different pathotypes of *V. dahliae* Kleb., defoliating and non-defoliating, which are named T1 and SS4, respectively, in the USA and in many parts of the world [12,13]. In Turkey, two deciduous and evergreen pathotypes were reported—the defoliating pathotype was seen in 93% of fields in the Aegean region, while the non-defoliating pathotype was seen in 77% of fields in the Eastern Mediterranean and Southeast Anatolia [14].

In breeding programs, it is crucial to develop resistant varieties to combat with the disease caused by this fungus [15]. In addition, the use of marker assisted selection (MAS) along with classical breeding methods in breeding disease-resistant varieties will increase breeding success [16].

In order to make marker assisted selection, it is necessary to determine the quantitative trait loci associated with the disease. Association mapping is widely used to determine quantitative trait loci [17]. The starting point of association maps is based on the non-random association of alleles at different loci [17].

With the advancing technology, the widespread use of sequencing technologies has increased the interest in single nucleotide polymorphism (SNP) markers and accelerated marker identification studies with association mapping. With the development of high capacity genotyping methods, SNPs have become even more attractive marker systems [18]. On the other hand, genotyping by sequencing (GBS) is accepted as a powerful genotyping method and its use is rapidly increasing in many plant species [19].

This study was carried out to determine the quantitative trait loci (QTL) associated with Verticillium wilt disease using SNP markers obtained by the GBS method and to enable marker assisted selection.

## 2. Results

Minor allele frequency (MAF) filtering was performed for 10,173 SNP data obtained from Beijing Genome Institute (BGI) to determine the markers related to Verticillium wilt disease and 4370 SNP markers used in association mapping analysis. In the field experiment, phenotyping was performed with Disease Severity Index (DSI) on the leaves at the 50–60% boll opening period and post-harvest DSI in the stem section, while in climate chamber Vd11 and PVYD6 isolates were used in phenotyping and the data (in 2019) published by Çelik et al. [20].

Marker trait relationship was determined through the general linear model (GLM) and mixed linear model (MLM). Once SNP markers for a character found in GLM analysis overlap with MLM, it increases the probability that these markers are the QTL we are looking for [21]. Therefore, both GLM and MLM methods were used to determine the marker trait relationship. Sequences of markers showing significant association are given in Appendix A.

### 2.1. Markers Associated with Resistance/Tolerance to Verticillium Wilt DSI on Leaves at the 50–60% Boll Opening Period

Ten markers related to disease resistance/tolerance were determined to be associated with DSI in the leaves at the 50–60% boll opening period. The determined markers were calculated as significant at the *p* ≤ 0.01 to *p* ≤ 0.0001 significance level by the GLM and at the *p* ≤ 0.01 to *p* ≤ 0.001 by the MLM. These markers explained the phenotypic variation (R²) of 14–29% and 14–28% by the GLM and the MLM, respectively (Table 1). The -LOG10 (*p*-Value) value of markers associated with Verticillium wilt for DSI on the leaves at the 50–60% boll opening period were determined to vary between 2.79 and 3.93 in the GLM and from 2.54 to 3.30 in the MLM (Figure 1).

It has been known that these markers are located on chromosomes 2, 3, 16, 19, 21, 22, 24, 25 and 26 of *G. hirsutum* L. and the D4, D9 and D11 chromosomes of the D genome [22] of cotton. Markers located on protein coding genes are marker A3190 on Protein Mpv17-like (LOC107948714), marker A5075 on 21 kDa protein-like (LOC107903729, LOC107921167), marker A4529 on transcription factor MYB113-like (LOC105804552), marker A2118 on protein dehydration-induced 19 homologous 3-like of (LOC107912355, LOC107924363), marker A5526 on LOC107941261, marker A8104 on extracellular ribonuclease LE-like (LOC107902274, LOC106419674), marker A464 on F-box protein CPR30-like (LOC107944641), and marker A9833 on putative E3 ubiquitin-protein ligase LIN (LOC107897477).

### 2.2. Post-Harvest DSI in Stem Section and Markers Associated with Verticillium Wilt Disease Resistance/Tolerance

Eight markers related to resistance/tolerance to Verticillium wilt DSI in the stem section were determined. Markers were significant at the *p* ≤ 0.01 to *p* ≤ 0.001 level according to the GLM method and at the *p* ≤ 0.01 significance level according to the MLM method. These markers were found to explain the phenotypic variation (R²) between 17–29% in the GLM and 15–30% in the MLM (Table 2). The -LOG10 (*p*-Value) value of the markers associated with the stem cross section of verticillium wilt varied between 2.77–3.63 and 1.91–3.02 for the GLM and MLM, respectively (Figure 2).

Moreover, these markers with significant relationships have already been shown to be located on chromosomes 6, 10, 12, 17 and 24 of *G. hirsutum* L. and the D3, D8 and D9 chromosomes of the D genome [22] of cotton by performing blast analysis. Markers A8451, A9046, A2768 and A4120 were located on genes that are protein coding LOC107963741-LOC107960135 genes, the pentatricopeptide repeat-containing protein At3g62890-like (LOC107899479) gene, the protein coding LOC107918458 gene, and protein coding chloroplastic-like (LOC107948854-LOC107899022), respectively.

### 2.3. Markers Associated with Non-Defoliating Pathotype (Vd11 Isolate) of Verticillium Wilt Disease

Twenty-one markers associated with Verticillium wilt non-defoliating (Vd11 isolate) pathotype were identified. The determined markers were significant at the *p* ≤ 0.01 to *p* ≤ 0.0001 level according to the GLM method and at the *p* ≤ 0.01 significance level according to the MLM method. These markers explained the phenotypic variation (R²) between 15–24% and 14–24% for the above-mentioned methods, respectively (Table 3). The -LOG10 (*p*-Value) value of the markers associated with Vd11 pathotype varied between 2.77–3.91 according to the GLM method and 2.04–3.28 according to the MLM method (Figure 3).

In addition, these markers with significant relationships were found on chromosomes 1, 4, 5, 10, 12, 15, 16, 21, 22, and 25 of *G. hirsutum* L. and the D1, D4, D5, D6, D9 and D10 chromosomes of the D genome of cotton through NCBI blast analysis [22]. On the other hand, related markers located on the protein coding genes were found to be as follows: marker A5502 on protein IRK-interacting protein-like, transcript variant X2 (LOC107911588), marker A4657 on glycosylphosphatidylinositol anchor attachment 1 protein (LOC105780137), A2168 marker on U3 small nucleolar RNA-associated protein 4-like (LOC107896835), marker A7267 on microtubule-associated protein futsch-like (LOC107946501), marker A5067 on transport and Golgi organization 2 homolog (LOC107921815), marker A1772 on protein networked 1D-like (LOC107951191), marker A6025 on splicing factor 3B subunit 3-like (LOC107959892), marker A680 on mediator of RNA polymerase II transcription subunit 15a-like (LOC107914147, LOC107897284) and marker A6147 on putative ankyrin repeat protein RF_0381, ankyrin-3-like (LOC107893150, LOC107937316).

### 2.4. Markers Associated with Defoliating Pathotype (PYDV6 Isolate) of Verticillium Wilt Disease

Twenty-three markers associated with the PYDV6 isolate were identified. Markers were calculated to be significant at the *p* ≤ 0.01 to *p* ≤ 0.0000 level by the GLM and at *p* ≤ 0.01 to *p* ≤ 0.001 by the MLM. These markers explained the phenotypic variation (R²) of 17–39% and 15–42% for the GLM and MLM, respectively (Table 4). The -LOG10 (*p*-Value) value of the markers associated with PYDV6 pathotype varied between 2.76–5.07 according to the GLM method and 2.03–4.06 according to the MLM method (Figure 4).

In addition, these markers were located on chromosomes 1, 3, 16, 18, and 21 of *G. hirsutum* L. and the D1, D2, D6, D7, D8, D11 and D13 chromosomes of the D genome after blast analysis of NCBI [22]. Additionally, markers on protein coding genes were determined as follows: marker A412 on tubby-like F-box protein 8, transcript variant X2 (LOC107925491), marker A4574 on endoglucanase 16-like (LOC107932833), marker A5855 on glucose-6-phosphate/phosphate translocator 2, chloroplastic-like (LOC107901936), marker A4939 on LOC107897225, marker A9364 on metal tolerance protein 11-like (LOC107897780), marker A968 on VAN3-binding protein-like (LOC107914851), marker A7740 on transformation/transcription domain-associated protein-like (LOC107912994), marker A1176 on pyruvate kinase isozyme A, chloroplastic-like (LOC107929115), marker A4948 on ethylene-responsive transcription factor CRF2-like (LOC107920268), marker A3067 on molybdate transporter 2-like (LOC107897224) and marker A1756 on non-functional NADPH-dependent codeinone reductase 2-like (LOC107933372).

## 3. Discussion

### 3.1. Markers Associated with Resistance/Tolerance to Verticillium Wilt DSI on Leaves at the 50–60% Boll Opening Period

Marker A3190, which we determined to have a significant relationship DSI on leaves at the 50–60% boll opening period, was located on the protein Mpv17-like (LOC107948714) gene, and it has been reported that Mpv17 contributes to osmotic stress tolerance in plants [23]. Considering the fact that Verticillium wilt restricts the water and substance transport of the plant by occluding the plant transmission bundles and thus increasing the osmotic pressure, our observation seems reasonable. In addition to this, the A5075 marker overlaps the 21 kDa protein-like gene. The 21 kDa protein-like gene is included in late embryogenesis-abundant (LEA) protein group 1 [24] and is known to be involved in stress tolerance [25]. It is thought that the marker A4529 overlaps with the transcription factor MYB113 gene, which is involved in the initiation of anthocyanin biosynthesis in the late period. This gene is protective and preventive against many pathogens [26,27], and this gene may be preventive in wilt stress. Marker A2118 is on the protein dehydration-induced 19 homologous 3-like gene, and this gene is expressed in high amounts as a protective seed embryo in vegetative parts in case of water deficiency [28]. It has been determined that the A8104 marker is located on the extracellular ribonuclease LE-like gene. It is stated that the extracellular ribonuclease LE-like gene is associated with resistance to fungi and pathogens [29]. Marker A464 was overlapped with the F-box protein CPR30-like gene. This gene provides structural resistance to the bacterial disease *Pseudomonas syringae* which causes dwarfism in Arabidopsis [30], and this gene may be similarly located in the defense mechanism against Verticillium wilt disease. With this, marker A9833 is on the putative E3 ubiquitin-protein ligase LIN gene, and this gene regulates the response to phytohormones (brassinolide, gibberellic acid (GA), indole-3-acetic acid (IAA), and salicylic acid (SA)) and abiotic stress (cold, heat, drought, and salt) [31]. The putative E3 ubiquitin-protein ligase LIN gene may also be effective in the regulation of mechanisms against stress factors, so it can be involved in regulation and develop as a response to Verticillium wilt disease. On the other hand, regarding Verticillium wilt resistance, QTL were determined on chromosomes 4, 9, and 11, with the majority on 16 [32], 4, 22, 24, and 26 [33], 26 and 21 [34]. It has been determined that the markers we have determined are on the same chromosomes previously determined by different researchers.

### 3.2. Post-Harvest DSI in the Stem Section and Markers Associated with Verticillium Wilt Disease Resistance/Tolerance

The A9046 marker, with its stem section associated with wilt intensity, was found on the pentatricopeptide repeat-containing protein At3g62890-like gene. Pentatricopeptide repeat protein genes in plants are one of the largest and most mysterious gene families [35] and transcriptional regulators of cytoplasmic genes [36]. The Pentatricopeptide repeat protein (At3g62890) gene, which we found to be associated with Verticillium wilt disease in cotton, may give researchers an idea about the function of this gene in the future. With this, the A4120 marker, with its stem section associated with the intensity of wilting, was on the fructose-1,6-bisphosphatase (LOC107948854 and LOC107899022) gene. Fructose-1,6-bisphosphatase (FBP) is a key enzyme in the plant sucrose synthesis pathway in the Calvin cycle and plays an important role in the regulation of photosynthesis in green plants. In addition, it is associated with the fiber quality of cotton as well as salt stress and Verticillium wilt disease [37]. Marker A8451, with which we found a very strong association in the 10th and 12th chromosomes of the cotton genome, was located on the uncharacterized protein LOC107963741-LOC107960135. It is clear that this previously uncharacterized gene is associated with Verticillium wilt disease.

### 3.3. Markers Associated with Non-Defoliating Pathotype (Vd11 isolate) of Verticillium Wilt Disease

Marker A5502, associated with the Vd11 isolate, was determined to be on the IRK-interacting protein (LOC107911588) gene. It has been reported that the IRK-interacting protein is involved in the maintenance and differentiation of the root and shoot meristem [38,39]. In the meantime, marker A4657 has been determined to be located on the glycosylphosphatidylinositol anchor attachment 1 protein gene, and glycosylphosphatidylinositol anchor genes play a key role in many biological processes by targeting proteins to the cell wall; however, its role in plant pathogenic fungi is largely unknown [40]. With this study, it was determined that the glycosylphosphatidylinositol anchor gene is associated with Verticillium wilt. Marker A7267 was located on the microtubule-associated protein futsch-like gene. In others, microtubule genes control aspects of cell division and expansion and are essential for plant morphogenesis and development. In addition, microtubules are associated with cellulose synthase complexes as well as the accumulation of cellulose micro-fibrils, particularly during the secondary wall deposition period, and direct cellulose micro-fibril accumulation [41,42]. *V. dahliae* pathogens attack the celluloses in the cotton plant’s transmission bundles, preventing water and nutrient transport to the plant. The microtubule-associated protein futsch-like gene supports the relationship we have identified due to its role in cellulose synthesis. However, marker A5067 was located on the transport and Golgi organization in the 2 homologous gene, and although this gene has been reported to be involved in traffic between the endoplasmic reticulum and the Golgi apparatus [43], its function has not been clarified. In this study, it has been determined that this gene has a relationship with Verticillium wilt disease. On the other hand, transport and Golgi organization in the 2 homologous gene have been reported to be associated with some diseases in humans and *drosophila* [43]. The A1772 marker was located on the LOC107951191 (protein networked 1D-like) gene and there is no information about the function of this gene. With this study, we propose that the protein networked 1D-like gene may be associated with Verticillium wilt disease. In the meantime, the A6025 marker was located on the splicing factor 3B subunit 3-like gene. Knowledge about the role of this gene in plants is limited. On the other hand, Chen et al. [44] reported that this gene is likely to play important roles in the regulation of cell cycle, proliferation and death in their study on rice. Marker A680 was determined to be on the mediator of RNA polymerase II transcription subunit 15a-like gene. It was reported that the transcription by RNA polymerase II (Pol II) enzyme occurs not only in the annotated protein-coding genes, but throughout the genome and is essentially important for most physiological processes [45]. Furthermore, Thieme et al. [46] reported that the RNA polymerase II (Pol II) gene may be associated with stress. The A6147 marker is on the putative ankyrin repeat protein gene and the putative ankyrin repeat protein family is widely distributed among plants and has been found to participate in multiple processes, such as plant growth and development, hormone response, and response to biotic and abiotic stresses [47]. More importantly, this gene has been reported to be required for disease resistance [48], and it has been found that the markers we have identified are on same chromosomes previously identified by different researchers [49].

### 3.4. Markers Associated with Defoliating Pathotype (PYDV6 Isolate) of Verticillium Wilt Disease

Marker A412 tubby-like F-box protein 8 was determined to be on the transcript variant X2 (LOC107925491) gene. Although Tubby-like protein genes have been known to play an important role in environmental stresses, their functions were rarely elucidated. However, there have been some studies where the genes were associated with disease tolerance in cotton and some other plants [50]. Apart from that, the 4574 marker was observed to be located on the endoglucanase 16-like (LOC107932833) gene. As a member of a family of cellulatic enzymes, endoglucanase plays an important role in cellulose degradation [51]. It is found in plants in abundance and weakens the cell walls of the pathogens attacking the plant [52]. On the other hand, marker A5855 was located on the glucose-6-phosphate/phosphate translocator 2 (LOC107901936) gene. It has been stated that this gene is involved in CO_2_ assimilation, photosynthesis and starch accumulation in plants [53,54,55,56]. However, its relationship with biotic or abiotic stress has not been established yet. This study reveals that the glucose-6-phosphate/phosphate translocator 2 gene can also be associated with Verticillium wilt disease. Another marker, A9364, was located on the metal tolerance protein 11-like (LOC107897780) gene. Although the metal tolerance protein 11 gene has been demonstrated to be associated with tolerance to metals in some plants, there is no information about its involvement in diseases in plants [57,58,59]. In this study, we suggested that the metal tolerance protein eleven gene may have a relationship with Verticillium wilt. Additionally, the A968 marker was found to be located on the VAN3-binding protein-like (LOC107914851) gene. Naramoto and Kyozuka [60] stated that cell polarity formation, cell wall formation, and biotic and abiotic responses in plants all depend on intracellular membrane traffic, and previous studies have reported that VAN3 is localized in the plasma membrane and intracellular structures. Here, we raised the possibility that the VAN3-binding protein-like gene may have a relationship with Verticillium wilt disease. Marker A4988 was located on the ethylene-responsive transcription factor CRF2-like (LOC107920268) gene. The ethylene-responsive transcription factor has been reported to develop a response to various biotic and abiotic stresses in some model plants [61,62,63]. These results support our findings. Lastly, marker A3067 was located on the molybdate transporter 2-like gene, and the transcription of this gene has been identified in aging leaves [64]. Verticillium wilt disease causes premature aging and leaf shedding. This is evidence that a relationship may exist with the molybdate transporter 2-like gene in wilt disease. In addition, it has been found that the markers we have identified are on same chromosomes previously identified by different researchers [65,66,67].

## 4. Materials and Methods

### 4.1. Plant Materials and Pathogen Isolates

The experimental study started with two thousand and eighty-four cotton genotypes (*G. hirsutum* L.) obtained from different parts of the world (Appendix A), but ninety selected genotypes as a result of disease testing were used in association mapping work. Disease pathotypes PYDV6 (defoliating pathotype) and Vd11 (non-defoliating pathotype) with known virulence isolated from susceptible varieties were used in artificial inoculation [20].

### 4.2. Phenotyping Analysis

Phenotyping consisted of two stages: growth chamber (artificial inoculum) and field conditions (natural inoculum). The field experiment was carried out in Diyarbakır (Bismil)/Turkey in 2016 in a farmer’s field, which is naturally contaminated by the non-defoliating pathotype of *V. dahliae*. The inoculum density of non-defoliation pathogen in this naturally infested field was determined as 16.5 microsclerotia g^−1^ [68].

In the augmented experimental design, each genotype was single-rowed with a length of 12 m of an inter row spacing of 70 cm and 20 cm intra row. When the plants in each plot reached the 50–60% boll opening period, foliar evaluations by considering chlorosis (yellowness) and necrosis in the leaves were scored as described by Tsror et al. [69] with a slight modification in the scale (0: no symptoms; 1: chlorosis in lower leaves; 2: moderate (30–50% of leaves) wilt with severe chlorosis; 3: moderate wilt and necrosis; 4: severe (more than 50% of leaves) wilt and necrosis; 5: dead plant).

At the end of the harvest, vascular discoloration was determined by cutting the plants in each plot at 10 cm above soil level and evaluated using the 0–4 scale (0: there is no browning in the vascular system; 1: very slight discoloration of the woody texture; 2: brownish pattern scattered throughout the woody texture; 3: dark brown staining on all sides of the woody texture; 4: intense uniform browning and deterioration of the woody texture) [70], according to the color changing of the xylem tissues.

The susceptibility of upland cotton genotypes to *V. dahliae* in the pot experiment was determined by the conidial suspension technique using a scale of 0 to 5 [20] in a growth chamber. DSI values were calculated and obtained data were subjected to Arcsin for transformation [71]. Phenotyping data have been published by Çelik et al. [20].

### 4.3. Genotyping Analysis

In DNA isolation, leaf samples were first put into 2 mL of Eppendorf tubes and ground within liquid nitrogen. The DNA isolation protocol developed by Zhang and Steward [72] and modified by Bardak and Bolek [73] was followed. DNA quality was controlled by Nanodrop 2000 (Thermo Fisher Scientific) and agarose gel electrophoresis.

A DNA library was constructed with the restriction enzyme ApeKI following the GBS protocol detailed by Elshire et al. [74]. Sequencing the DNA was performed using GBS [74,75], which was run by HiSeq 2000 in the Beijing Genome Institute (BGI). The BGI also completed SNP mining, map construction and sequence assemblies by using SOAP family software [76].

### 4.4. Association Analysis

Q-matrix data were obtained from the Structure 2.3.4 program, which modeled the Bayesian method to make associations in the TASSEL 5 software [77]. The K value, which is the number of the subpopulations, was chosen between 1 and 10. Therefore, each K was run 10 times. The Markov Chain Monte Carlo (MCMC) length of the burn-in period was arranged as to 10,000 and the number of MCMC iterations after the burn-in was arranged to 100,000. The result file was converted to a zip file and transferred to the Structure Harvester program, and the Q matrix was determined [78].

The relationship between the DNA markers was determined in the TASSEL 5 software based on the association mapping of Bradbury et al. [79]. The minor allele frequencies of the SNP markers (TASSEL 5 > Filter > Sites) less than 0.05 (MAF < 0.05) obtained by the GBS method were deleted, and then the K-matrix was obtained. Association mapping analysis was performed according to the general linear model (GLM) and mixed linear model (MLM) methods. In association analysis, phenotypic, Q-matrix and genotypic data were used in the GLM method, and Q-matrix, kinship, phenotypic and genotypic data were used in the MLM method [80].

## Figures and Tables

**Figure 1 plants-10-00306-f001:**
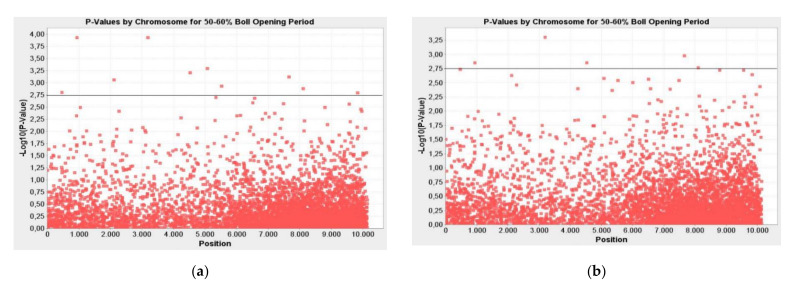
Manhattan plots for markers associated with resistance/tolerance to Verticillium wilt Disease Severity Index (DSI) on leaves at the 50–60% boll opening period; (**a**) general linear model (GLM); (**b**) mixed linear model (MLM); the line indicates the significance threshold (-LOG10 (*p*-Value) < 2.75).

**Figure 2 plants-10-00306-f002:**
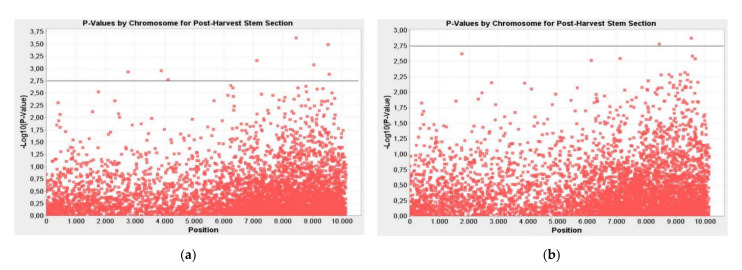
Manhattan plots for post-harvest stem section and markers associated with Verticillium wilt disease resistance/tolerance; (**a**) GLM; (**b**) MLM; the line indicates the significance threshold (-LOG10(*p*-Value) < 2.75).

**Figure 3 plants-10-00306-f003:**
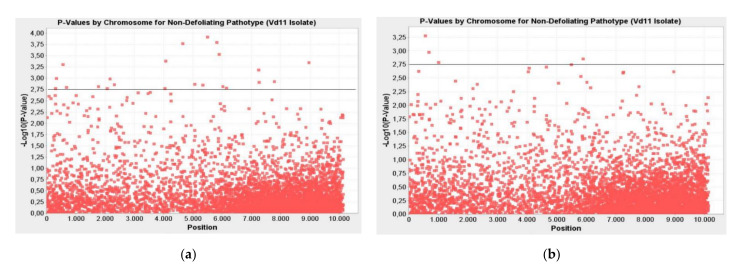
Manhattan plots for markers associated with non-defoliating pathotype (Vd11 isolate) of Verticillium wilt disease; (**a**) GLM; (**b**) MLM; the line indicates the significance threshold (-LOG10 (*p*-Value) < 2.75).

**Figure 4 plants-10-00306-f004:**
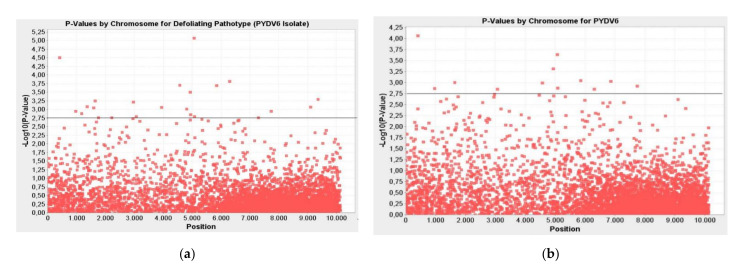
Manhattan plots for markers associated with defoliating pathotype (PYDV6 isolate) of Verticillium wilt disease; (**a**) GLM; (**b**) MLM; the line indicates the significance threshold (-LOG10 (*p*-Value) < 2.75).

**Table 1 plants-10-00306-t001:** Markers associated with resistance/tolerance to Verticillium wilt DSI on leaves at the 50–60% boll opening period.

Marker	Ch	Position	Gene	GLM			MLM		
				*p*-Value	R²	−LOG10 (*p*-Value)	*p*-Value	R²	−LOG10 (*p*-Value)
A3190	3	61.076.712	LOC107948714	0.0001	0.25	3.93	0.0005	0.25	3.30
A939	D11	42.444.312	NA	0.0001	0.29	3.93	0.0014	0.27	2.85
A5075	19 25	12.564.707 19.945.402	LOC107903729 LOC107921167	0.0005	0.25	3.30	0.0027	0.19	2.58
A4529	D11	36.723.905	LOC105804552	0.0006	0.22	3.21	0.0014	0.20	2.85
A7660	D9	29.949.760	NA	0.0008	0.26	3.12	0.0011	0.28	2.97
A2118	21 26	9.730.256 20.661.882	LOC107912355 LOC107924363	0.0009	0.25	3.06	0.0024	0.23	2.63
A5526	Na	3.027	LOC107941261	0.0012	0.18	2.93	0.0029	0.18	2.54
A8104	2 24	102.210.077 45.178.035	LOC107902274 LOC107919674	0.0013	0.22	2.88	0.0017	0.21	2.76
A464	D4	44.272.173	LOC107944641	0.0016	0.24	2.80	0.0018	0.23	2.74
A9833	2216	60.934.19163.975.517	LOC107897477	0.0016	0.14	2.79	0.0023	0.14	2.64

**Table 2 plants-10-00306-t002:** Post-harvest stem section and markers associated with Verticillium wilt disease resistance/tolerance.

Marker	Ch	Position	Gene	GLM			MLM		
				*p*-Value	R²	−LOG10 (*p*-Value)	*p*-Value	R²	−LOG10 (*p*-Value)
A8451	12 10	1.220.085 83.702.765	LOC107963741 LOC107960135	0.0002	0.21	3.63	0.0017	0.17	3.02
A9533	D8	48.778.681		0.0003	0.21	3.49	0.0013	0.18	2.73
A7122	D3	42.351.83		0.0007	0.27	3.17	0.0029	0.26	2.28
A9046	17	25.971.276	LOC107899479	0.0008	0.18	3.08	0.0087	0.11	2.62
A3886	D9	7.626.784		0.0011	0.29	2.95	0.0072	0.30	2.05
A2768	24	30.431.399	LOC107918458	0.0012	0.21	2.93	0.0070	0.17	2.04
A9566	D9	11.264.238		0.0013	0.17	2.88	0.0026	0.15	2.57
A4120	6 17	67.041.464 9.190.733	LOC107948854 LOC107899022	0.0017	0.24	2.77	0.0088	0.22	1.91

**Table 3 plants-10-00306-t003:** Markers associated with non-defoliating pathotype (Vd11 isolate) of Verticillium wilt disease.

Marker	Ch	Position	Gene	GLM			MLM		
				*p*-Value	R²	−LOG10 (*p*-Value)	*p*-Value	R²	−LOG10 (*p*-Value)
A5502	21 D11	2.450.321 3.328.322	LOC107911588	0.0001	0.20	3.91	0.0018	0.20	2.75
A5821	D5	29.389.195		0.0002	0.24	3.80	0.0029	0.24	2.53
A4657	12	9.125.748	LOC105780137	0.0002	0.19	3.77	0.0020	0.19	2.70
A5901	NA	NA		0.0003	0.18	3.53	0.0014	0.18	2.85
A4075	D4	13.047.156		0.0004	0.19	3.38	0.0021	0.19	2.68
A8965	D6	18.954.790		0.0004	0.18	3.35	0.0024	0.17	2.62
A558	D10	53.314.674		0.0005	0.23	3.30	0.0005	0.24	3.28
A7246	D4	42.119.775		0.0007	0.21	3.18	0.0026	0.21	2.59
A339	D6	27.507.643		0.0010	0.21	3.00	0.0024	0.21	2.63
A2168	16	24.453.514	LOC107896835	0.0010	0.19	2.98	0.0050	0.18	2.31
A7785	D4	158.221		0.0012	0.17	2.92	0.0045	0.16	2.34
A7267	5	4.645.398	LOC107946501	0.0012	0.19	2.91	0.0025	0.18	2.61
A5067	25	42.960.772	LOC107921815	0.0014	0.15	2.87	0.0039	0.16	2.41
A2316	D1	518.533.70		0.0014	0.19	2.86	0.0041	0.19	2.39
A5343	D9	39.774.356		0.0014	0.15	2.85	0.0092	0.14	2.04
A1772	1	87.736.674	LOC107951191	0.0015	0.18	2.81	0.0074	0.17	2.13
A6025	10	66.517.458	LOC107959892	0.0015	0.22	2.81	0.0038	0.22	2.43
A680	22 16	63.55.122 57.747.916	LOC107914147 LOC107897284	0.0016	0.20	2.80	0.0011	0.20	2.97
A6147	4 15	76.412.909 21.225.148	LOC107937316 LOC107893150	0.0017	0.16	2.78	0.0048	0.16	2.32
A4041	D6	13.120.530		0.0017	0.18	2.78	0.0024	0.18	2.61
A306	Mt	656.946		0.0017	0.16	2.77	0.0085	0.16	2.07

**Table 4 plants-10-00306-t004:** Markers associated with defoliating pathotype (PYDV6 isolate) of Verticillium wilt disease.

Marker	Ch	Position	Gene	GLM			MLM		
				*p*-Value	R²	−LOG10 (*p*-Value)	*p*-Value	R²	−LOG10 (*p*-Value)
A5072	D11	4.784.898		0.0000	0.34	5.07	0.0002	0.33	3.65
A412	D1	7.438.178	LOC107925491	0.0000	0.39	4.50	0.0001	0.42	4.06
A6306	D2	36.632.816		0.0002	0.23	3.81	0.0014	0.21	2.85
A4574	D8	61.214.324	LOC107932833	0.0002	0.27	3.70	0.0010	0.28	2.99
A5855	18	49.931.223	LOC107901936	0.0002	0.24	3.70	0.0009	0.21	3.04
A4939	16	56.101.995	LOC107897225	0.0003	0.27	3.50	0.0005	0.30	3.31
A9364	16	90.067.393	LOC107897780	0.0005	0.18	3.29	0.0039	0.13	2.42
A1641	D13	51.408.054		0.0006	0.31	3.25	0.0010	0.30	3.01
A2970	D7	33.833.694		0.0006	0.27	3.21	0.0019	0.24	2.73
A1365	NA	NA		0.0008	0.24	3.08	0.0024	0.25	2.63
A9109	D2	46.908.558		0.0009	0.17	3.07	0.0024	0.15	2.62
A3946	NA	NA		0.0009	0.23	3.06	0.0093	0.19	2.03
A1595	NA	NA		0.0009	0.28	3.04	0.0032	0.25	2.50
A4814	D9	20.084.101		0.0010	0.23	3.01	0.0026	0.20	2.59
A968	3	27.667.043	LOC107914851	0.0011	0.27	2.95	0.0014	0.30	2.86
A7740	D11 21	13.160.961 21.368.426	LOC107912994	0.0011	0.19	2.95	0.0012	0.21	2.92
A1176	NA	NA	LOC107929115	0.0013	0.26	2.88	0.0027	0.26	2.58
A4948	1	27.108.899	LOC107920268	0.0014	0.21	2.85	0.0020	0.22	2.70
A5085	NA	NA		0.0016	0.20	2.80	0.0013	0.22	2.87
A3067	16	56.089.930	LOC107897224	0.0016	0.20	2.79	0.0014	0.22	2.85
A2219	D6	25.619.204		0.0017	0.18	2.76	0.0081	0.17	2.09
A7299	D7	13.842.749		0.0017	0.17	2.76	0.0028	0.18	2.55
A1756	NA	303.898	LOC107933372	0.0018	0.22	2.76	0.0021	0.24	2.68

## Data Availability

Representations of the data are contained in the article, raw data is available upon request.

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
