# Peer review of "Association Mapping of Verticillium Wilt Disease in a Worldwide Collection of Cotton (Gossypium hirsutum L.)"

_plants, 2021, doi:10.3390/plants10020306_

Round 1
Reviewer 1 Report
Reviewer: 1
Ms. Ref. No.: Plants-1103458
Authors: Adem Bardak, Sadettin Celik, Oktay Erdogan, Remzi Ekinci and Ziya Dumlupinar.
Specific notes:
TITLE
Nothing to comment.
ABSTRACT
Line 17: Delete “we” and rewrite the sentence.
Line 19: Delete the significance (p <0.01).
KEYWORD
Nothing to comment.
INTRODUCTION
Lines 43 and 44: Change “Verticillium dahliae”, must appear in italics.
General comment to the introduction: It is a bit brief; I would suggest the authors add more details about the development of the disease in the crop and the use of the proposed methodology in other crops.
RESULTS
Line 170: Table 2?, it not should be table 4 ?, please check and correct. There is also a table 2 on line 119, but I think this table does not refer to a continuation of table 2
DISCUSSIÓN
Lines 235 and 236: “Verticillium dahliae”, must be abbreviated.
Line 245: dorosiphila?, please correct.
MATERIALS AND METHODS
Nothing to comment.
TABLES
Nothing to comment.
FIGURES
Figure 1: I suggest that the authors increase the size of the font that is on the axes of both graphs. Its small size makes it impossible to read.
Figure 2: as described above.
Figure 3: as described above.
Figure 4: as described above.
Author Response
Dear Reviewer,
The corrected manuscript is attached. Please see the attachment.
Line 17: Deleted “we” and the sentence was rewritten.
Deleted the significance (p <0.01).
More details about the development of the disease in the crop and the use of the proposed methodology in other crops was added in the introduction.
Table 4 corrected in line 179.
"Verticillium dahliae" in line 253 and236 is abbreviated.
Phenotypic analysis in materials and methods was reorganized and Disease Severity Index (DSI) of the leaves at the 50-60% boll opening period has been written in other parts of the manuscript.
Font size of all figures is increased.
Genotype origins was checked and corrected in supplementary Table 2.
Reference list reorganized.
Thank you for your contribution.
Best regards.

Reviewer 2 Report
This paper verifies findings of many others that Verticillium wilt disease in cotton is complex, difficult to characterize genetically, and, although a biotic stress, its phenotypic response is associated with quantitative abiotic stress tolerance.
- Objective is clear and concise.
- There is something missing in the third sentence of abstract. The message is developing resistant varieties is the most economical and practical way to manage Verticillium wilt in cotton after cultural practices since there is no chemical control. I think the word 'chemicals' might be missing, but there is a better way to express the meaning and need for tools to facilitate breeding resistant varieties.
- Throughout the paper, starting in the abstract, method for phenotyping is unclear. "50-60% boll opening period in leaves" makes no sense. Citation 58 (Erdogan 2009) is meant to indicate phenotyping method, but full text is not accessible. Both in methods section and in subsequent discussion, clarify that Verticillium wilt ratings in leaves and stem were taken at 50-60% open boll stage in the field. It is important to distinguish the phenotyping from other methods that might indicate percent defoliation of percent incidence. This was in a related paper by same authors and presumably is the correct method:When plants reached about 50-60% boll opening period, all the plants in the two middle rows of every parcel were screened for wilt disease symptoms on the leaves using 0-4 wilt scale (0 = no symptoms; 1 = 1-33% foliage affected; 2 = 34-66% foliage affected; 3 = 67-100% foliage affected; 4 = dead plants) (BejaranoAlcazar et al., 1995). In order to determine the stem section based wilt disease severity all mentioned plants were cut at 10 cm above the ground level (after harvest). The plant cuts were applied to color changes of vascular system based of the 0-3 scale (0 = no discoloration xylem on trunk sectional area; 1 = 1-33% discoloration xylem; 2 = 34-67% discoloration xylem; 3 = 68-100% discoloration xylem) (Buchenauer and Erwin, 1976). Disease rates were calculated and obtained data were subjected to Arcsin for transformation (Karman, 1971).
- Section 2.1 "Ten markers related to disease resistance/tolerance were determined in leaves during the 50-60% boll opening period." should be improved to adequately describe the phenotypic association, ie., "Ten markers related to disease resistance/tolerance were determined to be associated with disease symptoms in the leaves measured at 50-60% boll opening period."
- Citation #61 in the text (line 320) should be Zhang and Stewart (not Zang and Steward)
- The slate of germplasm selected for the association appears robust for G. hirsutum. Carmen is still probably one of the best germplasm lines for Verticillium wilt tolerance. Some PI numbers were checked for duplication (none found). 'G. hirsutum' of USA origin could be many things. It is a testament to global germplasm utilization that 'Flora' of Turkey origin is the same variety as 'Fibermax 958' of USA origin (and actually originated in Australia. Others listed as USA origin such as Siokra L22 (Siokra L23?) are likely Australian origin, too.
Author Response

(The authors gave the same response as above.)
